# Error-based or target-based? A unified framework for learning in recurrent spiking networks

**Cristiano Capone** [1]☯*, **Paolo Muratore** [2]☯*, **Pier Stanislao Paolucci** [1]

**1** INFN, Sezione di Roma, Rome, Italy, **2** Cognitive Neuroscience, SISSA, Trieste, Italy

☯ These authors contributed equally to this work.
* cristiano0capone@gmail.com (CC); pmurator@sissa.it (PM)

**Data Availability Statement:** The code associated to this paper is made publicly available in the following repository: https://github.com/myscience/goal. We provide two Python

## Abstract

The field of recurrent neural networks is over-populated by a variety of proposed learning rules and protocols. The scope of this work is to define a generalized framework, to move a step forward towards the unification of this fragmented scenario. In the field of supervised learning, two opposite approaches stand out, error-based and target-based. This duality gave rise to a scientific debate on which learning framework is the most likely to be implemented in biological networks of neurons. Moreover, the existence of spikes raises the question of whether the coding of information is rate-based or spike-based. To face these questions, we proposed a learning model with two main parameters, the rank of the feedback learning matrix R and the tolerance to spike timing $\tau_\star$. We demonstrate that a low (high) rank R accounts for an error-based (target-based) learning rule, while high (low) tolerance to spike timing promotes rate-based (spike-based) coding. We show that in a store and recall task, high-ranks allow for lower MSE values, while low-ranks enable a faster convergence. Our framework naturally lends itself to Behavioral Cloning and allows for efficiently solving relevant closed-loop tasks, investigating what parameters $(R, \tau_\star)$ are optimal to solve a specific task. We found that a high R is essential for tasks that require retaining memory for a long time (Button and Food). On the other hand, this is not relevant for a motor task (the 2D Bipedal Walker). In this case, we find that precise spike-based coding enables optimal performances. Finally, we show that our theoretical formulation allows for defining protocols to estimate the rank of the feedback error in biological networks. We release a PyTorch implementation of our model supporting GPU parallelization.

## Author summary

Learning in biological or artificial networks means changing the laws governing the network dynamics in order to better behave in a specific situation. However, there exists no consensus on what rules regulate learning in biological systems. To face these questions, we propose a novel theoretical formulation for learning with two main parameters, the number of learning constraints (R) and the tolerance to spike timing ($\tau_\star$). We demonstrate

implementations: a pure NumPy-based version and a PyTorch implementation with GPU support.

**Funding:** This work has been supported by the European Union Horizon 2020 Research and Innovation program under the FET Flagship Human Brain Project (grant agreement SGA3 n. 945539, to P.S.P., and grant agreement SGA2 n. 785907, to P. S.P.) and by the INFN APE Parallel/Distributed Computing laboratory as salary to P.S.P. C.C. received salary from SGA3 n. 945539 and SGA2 n. 785907. The funders had no role in study design, data collection and analysis, decision to publish, or preparation of the manuscript.

**Competing interests:** The authors have declared that no competing interests exist.

that a low (high) rank $R$ accounts for an error-based (target-based) learning rule, while high (low) tolerance to spike timing $\tau_\star$ promotes rate-based (spike-based) coding.

Our approach naturally lends itself to Imitation Learning (and Behavioral Cloning in particular) and we apply it to solve relevant closed-loop tasks such as the button-and-food task, and the 2D Bipedal Walker. The button-and-food is a navigation task that requires retaining a long-term memory, and benefits from a high $R$. On the other hand, the 2D Bipedal Walker is a motor task and benefits from a low $\tau_\star$.

Finally, we show that our theoretical formulation suggests protocols to deduce the structure of learning feedback in biological networks.

## Introduction

When confronted with reality, humans learn with high sample efficiency, benefiting from the fabric of society and its abundance of experts in relevant domains. A conceptually simple and effective strategy for learning in this social context is Imitation Learning. One can conceptualize this learning strategy in the Behavioral Cloning framework, where an agent observes a near optimal behavior (expert demonstration), and progressively improves its mimicking performances by minimizing the differences between its own and the expert's behavior. Behavioral Cloning can be directly implemented in a supervised learning framework. In the last years, a competition between two opposite interpretations of supervised learning is emerging: error-based approaches [1–5], where the error information computed at the environment level is injected into the network and used to improve later performances, and target-based approaches [6–13], where a target for the internal activity is selected and learned. In this work, we provide a general framework, which we call GOAL (Generalized Optimization of Apprenticeship Learning), where these different approaches are reconciled and can be retrieved via a proper definition of the error propagation structure the agent receives from the environment. Target-based and error-based are particular cases of our comprehensive framework. This novel formulation, being more general, offers new insights on the importance of the feedback structure for network learning dynamics, a still under-explored degree of freedom. Moreover, we remark that spike-timing-based neural codes are experimentally suggested to be important in several brain systems [14–17]. This evidence led us to we investigate the role of coding with specific patterns of spikes by introducing a parameter that defines the tolerance to precise spike timing during learning. Although many studies have approached learning in feedforward [9, 18–22] and recurrent spiking networks [2, 3, 8, 10, 12, 23, 24], a very small number of them successfully faced real world problems and reinforcement learning tasks [3, 25]. In this work, we apply our framework to the problem of behavioral cloning in recurrent spiking networks and show how it produces valid solutions for relevant tasks (button-and-food and the 2D Bipedal Walker). From a biological point of view, we focus on a novel route opened by such a framework: the exploration of what feedback strategy is actually implemented by biological networks and in the different brain areas. We propose an experimental measure that can help elucidate the error propagation structure of biological agents, offering an initial step in a potentially fruitful insight-cloning of naturally evolved learning expertise.

## Models

### The spiking network

In our formalism, neurons are modeled as real-valued variable $v_j^t \in \mathbb{R}$, where the $j \in \{1, \ldots, N\}$ label identifies the neuron and $t \in \{1, \ldots, T\}$ is a discrete time variable. Each neuron exposes

an observable state $s_j^t \in \{0, 1\}$, which represents the occurrence of a spike from neuron $j$ at time $t$. We then define the following dynamics for our model:

$$\hat{s}_i^t = \exp\left(-\frac{\Delta t}{\tau_s}\right)\hat{s}_i^{t-1} + \left(1 - \exp\left(-\frac{\Delta t}{\tau_s}\right)\right) s_i^t \tag{1}$$

$$v_i^t = \exp\left(-\frac{\Delta t}{\tau_m}\right)v_i^{t-1} + \left(1 - \exp\left(-\frac{\Delta t}{\tau_m}\right)\right)\left(\sum_j w_{ij}\hat{s}_j^{t-1} + I_i^t + v_{\text{rest}}\right) - w_{\text{res}}s_i^{t-1} \tag{2}$$

$$s_i^{t+1} = \Theta[v_i^t - v^{\text{th}}] \tag{3}$$

$\Delta t$ is the discrete time-integration step, while $\tau_s$ and $\tau_m$ are respectively the spike-filtering time constant and the temporal membrane constant. Each neuron is a leaky integrator with a recurrent filtered input obtained via a synaptic matrix $w \in \mathbb{R}^{N \times N}$ and an external signal $I_i^t$. $w_{\text{res}} = -20$ accounts for the reset of the membrane potential after the emission of a spike. $v^{\text{th}} = 0$ and $v_{\text{rest}} = -4$ are the threshold and the rest membrane potential.

## The supervised learning rule

We aim at training the recurrent spiking network to reproduce a desired output. In the framework of behavioral cloning, this output is the behavior of an expert agent (human, pre-trained artificial intelligence) which already knows an almost optimal solution of a task (see details in the section Application to closed-loop tasks: Behavioral cloning).

In order to train the network to reproduce at each time the desired output vector $y_k^{\star t}$, it is necessary to minimize the loss function:

$$\mathrm{E} = \sum_{t,k}(y_k^{\star t} - y_k^t)^2. \tag{4}$$

where $y_k^t = \sum_i \mathsf{B}_{ik}\bar{s}_i^t$ is a linear readout of the spiking activity of the network and $\mathsf{B}_{ik} \in \mathbb{R}$. $\bar{s}_i^t$ is defined as:

$$\bar{s}_i^t = \exp\left(-\frac{\Delta t}{\tau_\star}\right)\bar{s}_i^{t-1} + \left(1 - \exp\left(-\frac{\Delta t}{\tau_\star}\right)\right) s_i^t \tag{5}$$

i.e., a temporal filtering of the spikes $s_i^t$, where $\Delta t$ is the temporal bin of the simulation and $\tau_\star$ the timescale of the filtering.

It is possible to derive the learning rule by differentiating the previous error function (by following the gradient), similarly to what was done in [3]:

$$\Delta w_{ij} \propto \frac{d\mathrm{E}}{dw_{ij}} \simeq \sum_t \left[\sum_k \mathsf{B}_{ik}(y_k^{\star t+1} - y_k^{t+1})\right] p_i^t e_j^t, \tag{6}$$

where we have used $p_i^t$ for the pseudo-derivative (similarly to [3]) and reserved $e_j^t = \frac{\partial v_i^t}{\partial w_{ij}}$ for the spike response function that can be computed iteratively as

$$\frac{\partial v_i^{t+1}}{\partial w_{ij}} = \exp\left(-\frac{\Delta t}{\tau_m}\right)\frac{\partial v_i^t}{\partial w_{ij}} + \left(1 - \exp\left(-\frac{\Delta t}{\tau_m}\right)\right)\hat{s}_i^t. \tag{7}$$

In our case the pseudo-derivative, whose purpose is to replace $\frac{\partial f(s_i^{t+1}|v_i^t)}{\partial v_i^t}$ (since $f(\cdot)$ is non-differentiable, see Eq (3)), is defined as $p_i^t = \frac{e^{v_i^t/\delta v}}{\delta v(e^{v_i^t/\delta v}+1)^2}$, it peaks at $v_i^t = 0$ and $\delta v$ is a parameter defining its width. For the complete derivation, we refer to Section A in S1 Text (where we also discuss the $\simeq$ in Eq (6)).

## Results

In the following sections, we define a generalized learning framework by identifying two sensitive parameters: the number of constraints $R$ and the sensitivity to precise temporal coding $\tau_\star$. We analytically show how different learning rules presented in the literature can be accounted for as specific cases of our framework. We provide a geometrical interpretation, and strengthen our statement through numerical experiments.

Finally, we test the performances of our learning rule as a function of the two main parameters $(R, \tau_\star)$ on different tasks: a store and recall task (of a target 3D trajectory), a navigation task (button and food) and a motor task (2D bipedal walker).

### Theoretical results

**Generalization of the learning framework.**    As discussed above, in Eq (6) we defined $y_k^{\star t}$ as the desired output (e.g., the target behavior). However, it is possible to imagine that in both biological and artificial systems there are much more constraints, not directly related to the behavior, to be satisfied. One example is the following: it might be necessary for the network to encode an internal state which is useful to produce the behavior $y_k^{\star t}$ and to solve the task (e.g. an internal representation of the position of the agent, contextual information and so on). The encoding of this information can automatically emerge during training, however to directly suggest it to the network might significantly facilitate the learning process. This signal is referred to as hint in the literature [23]. For this reason, we introduce a further set of output targets $q_k^{\star t}, k = O + 1, \ldots R$ and define $Y_k^{\star t}, k = 1, \ldots R$ as the collection of $y_k^{\star t}$ and $q_k^{\star t}$. $Y_k^t$ is the signal decoded from the network activity through a linear readout $Y_k^t = \sum_i \mathsf{B}_{ki}^+ \bar{s}_i^t$ and it is constrained to be as similar as possible to the target $Y_k^{\star t}$ (see Fig 1A). By definition, $\mathsf{B}_{ki}^+$ is the same as $\mathsf{B}_{ki}$ but with $\mathsf{R} - \mathsf{O}$ extra rows. See section Definition of the additional constraints for details on the choice of $Y_k^{\star t}$ and $\mathsf{B}_{ki}^+$.

The gradient-based minimization of the error $\mathrm{E} = \sum_{k,t} (Y_k^{\star t} - Y_k^t)^2$ results in the following learning rule:

$$\Delta w_{ij} = \eta \sum_t \left[ \sum_k \mathsf{B}_{ki}^+ (Y_k^{\star t+1} - Y_k^{t+1}) \right] p_i^t e_j^t. \tag{8}$$

The possibility to broadcast specific local errors in biological networks has been debated for a long time [26, 27]. On the other hand, the propagation of a target appears to be more coherent with biological observations [28–31]. For this reason, we propose an alternative formulation allowing to propagate targets rather than errors [6, 27]. This can be easily done by writing the target output as:

$$Y_k^{\star t} \simeq \sum_i \mathsf{B}_{ki}^+ \bar{s}_i^{\star t}. \tag{9}$$

We stress here the fact that, due to the spikes discretization, the last equality cannot be strictly achieved, and it is only an approximation. One can simply consider $s^{\star t}$ to be the

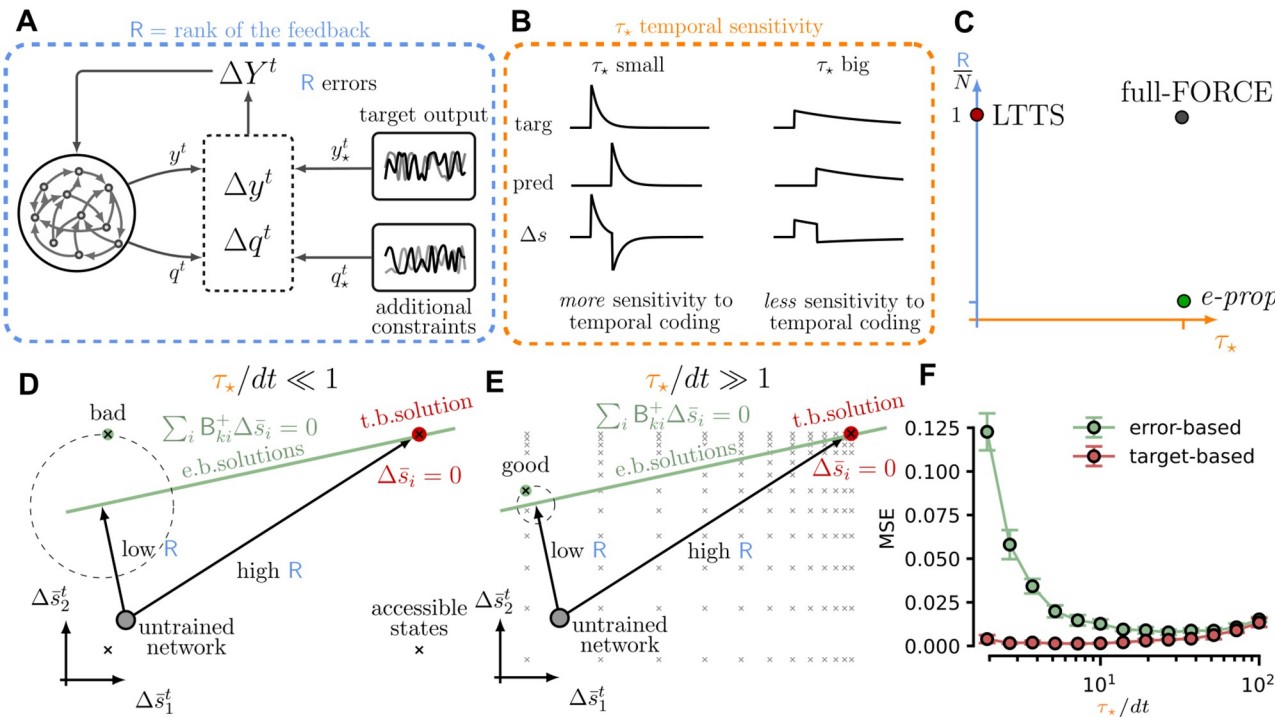

**Fig 1. Framework schematics.** We propose a general framework where the dimensionality of the error feedback $R$, and the sensitivity to temporal coding $\tau_\star$ can be varied arbitrarily. (**A**) Graphical depiction of a general supervised learning framework. $R$ errors (difference between the output and the target output) are projected to the network to evaluate the recurrent weights updates. (**B**) Graphical depiction of the role of the temporal sensitivity parameter $\tau_\star$. (**C**) Several learning rules present in the literature (e-prop, full-FORCE, LTTS) can be accounted as specific cases of our general framework. (**D**) Target-based approaches define a specific internal solution for network dynamics (red point), while error-based solutions are distributed in a subspace of possible internal states (green line). However, not all the points on the green line are accessible (given the discrete nature of the spiking activity), and the error-based solution can be sub-optimal. (**E**) However, when $\tau_\star$ is large, the accessible states are more dense, and it is easier to find a good solution with an error-based approach. (**F**) The MSE for target-based and error-based solutions becomes comparable for large $\tau_\star$ values. Points are average between five realization of the minimum MSE (after $10^4$ training epochs), with error bars denoting the corresponding standard deviations.

solution of the optimization problem $s_i^{\star t} = \mathrm{argmin}_{s_i^{\star t}} \sum_{kt} |y_k^{\star t} - \sum_i \mathsf{B}_{ki}^+ \bar{s}_i^{\star t}|$. The optimal encoding for a continuous trajectory through a pattern of spikes has been broadly discussed in [32]. However, the pattern $s^{\star t}$ might describe an impossible dynamics (for example, activity that follows periods of complete network silence). For this reason, here we take a different choice. The $s_i^{\star t}$ is the pattern of spikes expressed by the untrained network (where recurrent connections are all set to zero) when the target output $y_i^{\star t}$ is randomly projected as an input (similarly to [8, 9]). It has been demonstrated that this choice allows for fast convergence and encodes detailed information about the target output. With these additional considerations, we can now rewrite our expression for the weight update in terms of the network activity:

$$\Delta w_{ij} = \eta \sum_t \left[ \sum_k \mathsf{D}_{ik} (\bar{s}_k^{\star t+1} - \bar{s}_k^{t+1}) \right] p_i^t e_j^t, \qquad (10)$$

where $\mathsf{D} = \mathsf{B}^{+^\top} \mathsf{B}^+$ is a novel matrix which acts recurrently on the network. The two core new terms are the matrix $\mathsf{D}$ and $\bar{s}_k^{\star t+1}$. The former defines the dynamics in the space of the internal network activities $s_k^t$ during learning.

The latter provides a specific pattern of spikes, which is directly suggested to the network as the internal solution of the task. We interpret the parameter $\tau_\star$ (the time-scale of the spike filtering, see Eq (5)) as the tolerance to spike timing of the proposed internal solution $\bar{s}_i^{\star t}$. This clarifies the use of the subscript $\star$ for the timescale $\tau_\star$, since it concerns the target quantities. In Fig 1B we show in a sketch that, for the same spike displacement between the spontaneous and the target activity, the error is higher when the $\tau_\star$ is lower. However, as demonstrated in the following sections, the network dynamics only converges to $\bar{s}_k^{\star t+1}$ for a range of parameters R and $\tau_\star$.

**Definition of the additional constraints.**   As discussed above there are many possible choices for the additional constraints $q_k^{\star t}$ (contextual signals, f.r. regularization and so on), and we take the following one. We first compute the target spiking pattern $\bar{s}_k^{\star t}$ (as described in the section 2.2.1) and trained the network readout $B_{ik}$ (via standard output-error gradient descent, Adam optimizer) to reconstruct $y_i^{\star t} = \sum_k B_{ik} \bar{s}_k^{\star t}$. We remark that $\bar{s}_k^{\star t}$ only serves to define the readout weights, and it will not be used explicitly when using the learning rule in Eq (8).

Then, we define $Y_k^{\star t} = \sum_i B_{ki}^+ \bar{s}_i^{\star t}$. The first $O$ rows of the matrix $B_{ki}^+$ are taken from the matrix $B_{ki}$ so that $Y_k^{\star t} = y_k^{\star t}$ for $k \leq O$. The other rows are chosen randomly from a Gaussian distribution (with the same variance of the matrix $B_{ki}$). In summary, the generalized target $Y_k^{\star t}$ is:

$$Y_k^{\star t} = \begin{cases} y_k^{\star t} & \text{for } k \leq O \\[2mm] q_k^{\star t} & \text{for } O < k \leq R \end{cases} \tag{11}$$

where the additional constraints $q_k^{\star t}$ are a random linear combination of a hypothetical internal solution $\bar{s}_k^{\star t}$. However, we demonstrate that only when the rank is high, the internal dynamics converges to $\bar{s}_k^{\star t}$.

**Training protocol.**   We recap in this section the network training procedure used for rank-feedback control (Figs 1 and 2). First, we construct the target spike-pattern $s_i^{\star t}$ from the target output signal: the output signal is randomly projected into the untrained network, alongside the regular input. The spike activity expressed by the untrained network with such input is selected as internal target $s_i^{\star t}$. Then, we train the linear readout $B_{ki}$ to reproduce the target signal from the newly constructed target spike pattern. Note how, at this stage $B_{ki} \in \mathbb{R}^{O \times N}$. The output connectivity matrix $B_{ki}$ is then expanded to include new constraints, resulting in $B_{ki}^+ \in \mathbb{R}^{R \times N}$. In practice, we extract the $R - O$ new vectors from a Gaussian distribution with zero mean and variance equal to $2 \, \text{std}(B_{ki})$. Finally, the expanded output connectivity matrix $B_{ki}^+$ is adopted in the recurrent synaptic updates and Eq (8) is used for training.

**The target-based limit.**   We remark that the formulation described above is equivalent to an error-based approach whose output target is $Y_k^{\star t} \simeq \sum_i B_{ki}^+ \bar{s}_i^{\star t}$. When the rank of the matrix $B^+$ is comparable to the number of neurons, the matrix D is almost diagonal and the learning rule reduces to:

$$\Delta w_{ij} \simeq \eta \sum_t (\bar{s}_i^{\star t+1} - \bar{s}_i^{t+1}) p_i^t e_j^t. \tag{12}$$

In this case, the training of recurrent weights reduces to learning a specific pattern of spikes [33–37]. In this limit, the model Learning Through Target Spikes (LTTS) [9] is recovered (see Fig 1C), with the only difference of the presence of the pseudo-derivative. These two limits are investigated numerically in the section Dimensionality of the solution space. We remark that in the formulation described in Eq (10) it is possible to change the rank of the feedback by directly changing the number of rows in the matrix $B_{ik}^+$.

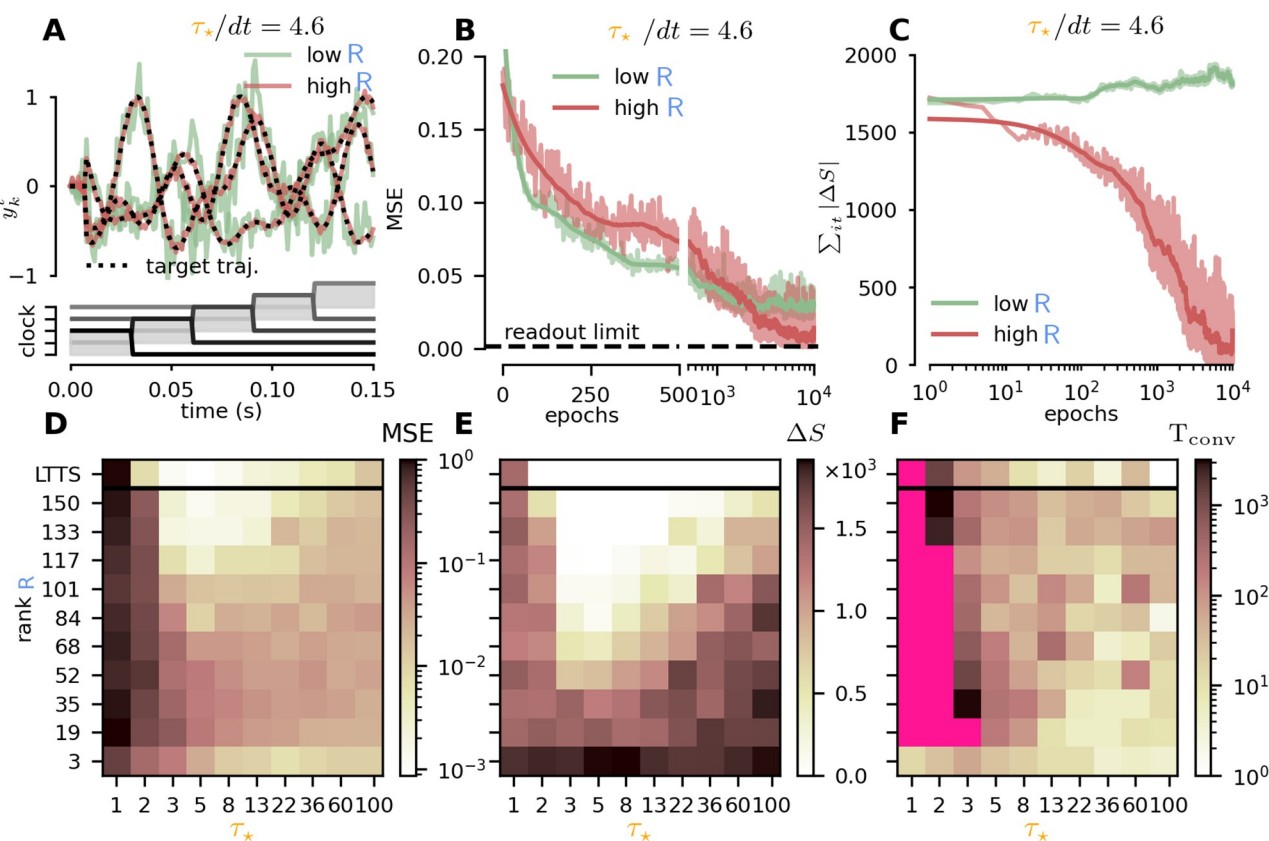

**Fig 2. Parameter exploration.** (**A**) We benchmarked our framework on a store and recall task. The network has to autonomously reproduce a target 3D trajectory, given a clock like input (bottom). Dashed line: target output. Solid line: network output at the end of the training for low and high rank conditions (green and blue respectively). (**B**) MSE (between the target output and the network output) as a function of training epochs, in the store and recall task of a 3D trajectory. Low rank (R = 3, green) and high rank (R = N, red) performances are compared. (**C**) Spike error $\sum_{it}|s_i^{*t} - s_i^t|$ as a function of training epochs, for high and low rank learning rule. (**D**) The MSE (color-coded) as a function of the rank R and the timescale $\tau_\star$. (**E**) The spike error (color-coded) as a function of the rank R and the timescale $\tau_\star$. (**F**) Convergence time (color-coded) as a function of the rank R and the timescale $\tau_\star$.

Numerical results confirming this theoretical prediction are reported in the following section. A major advantage of target-based limit is in the implementability and plausibility of the error propagation. In the general error-based case, the performances/activity of neurons has to be read, compared with the target output and then broadcasted to all neurons. This process requires time. This is reflected in the non locality in Eq (10) due to the matrix D. To update the weights between neuron $i$ and neuron $j$ it is necessary to know the local errors $\bar{s}_h^{*t} - \bar{s}_h^t$ from all other neurons. On the contrary, the rule derived in the target-based limit, only requires the local error of the post-synaptic neuron.

**Interpretation of the framework.** The major strength of our formulation is the capability to encompass very different learning approaches in the same framework. We have already noted in the previous section how in the high-rank and small-$\tau_\star$ limit, one recovers the LTTS model, where a specific pattern of spikes is learned. In the large-$\tau_\star$ regime the precise temporal-coding of spikes is blurred out (see Fig 1B), preventing learning for a specific spike-pattern. However, with a high-rank configuration a target rate-based internal solution is still identified: this is the known full-FORCE solution [8], where the learnt input currents induce an internal target activity, which is suited for the task. Loosening the learning constraints, i.e. reducing the feedback matrix rank R, progressively enlarges the space of internal solutions. When R

matches the output dimensionality, we recover the known error-based approach (e.g. e-prop [3]): no internal dynamics is imposed on the system, which is only guided by the projection of the output error. Fig 1C visually represents where these models are located in the $(\mathsf{R}, \tau_\star)$ plane. Our novel general description of different learning approaches offers a new tool to better investigate their relationships. Here we provide a geometrical interpretation of our model, that intuitively explains what is the role of the two parameters $\mathsf{R}$ and $\tau_\star$ in defining the network internal solution to a task. Fig 1D–1E represent the error space, the difference between network activity and the target activity at a specific time $t$ (two sample neurons are represented, $\Delta \bar{s}_1^t$ and $\Delta \bar{s}_2^t$). As discussed in the previous section, the target-based learning rule univocally defines one solution $\bar{s}_i^{\star t} = \bar{s}_i^t$, $\forall i, t$ ($\Delta \bar{s}_i^t = 0$, represented by the red point in Fig 1D–1E). Thus, the high-rank learning rule (and the target-based one) tries to make the network dynamics converge to the red point (as represented by the black arrow in Fig 1D–1E).

On the other hand, the error-based rule defines a set of possible solutions, defined by $\sum_i \mathsf{B}_{ki}^+ (\bar{s}_i^{\star t} - \bar{s}_i^t) \simeq 0$, $\forall k, t$ (this can be represented as a line in the space in Fig 1D–1E, green line) in which the MSE is low. In other words, using the low-rank learning rule is equivalent to looking for the closest solution next to the green line (in Fig 1D–1E) (as represented by the black arrow in Fig 1D–1E). However, not all the points on the green line are accessible to the network (given the discrete nature of the spiking activity, see Fig 1D black crosses), and the error-based solution achieved during the learning procedure can be sub-optimal (see Fig 1D, green point which is far from the green line). Indeed, when $\tau_\star$ is small $\bar{s}_i^t \simeq s_i^t$, and it can assume only values −1, 0, or 1. However, when $\tau_\star$ is large, the $s_i^t$ signals produced by the network are filtered, and the possible values for $\bar{s}_i^t$ are no longer −1, 0, or 1. As a result, the accessible states are denser in the space of possible internal solutions (see Fig 1E, black crosses), and it is easier to find a good solution with an error-based approach (see Fig 1E, green point, which is now closer to the green line). This theoretical prediction is confirmed in numerical experiments (see the following section).

## Numerical results

**Store and recall.** To investigate the role of the $\mathsf{R}$ and $\tau_\star$ parameters in our learning rule (see Eqs (8) and (10)), we considered a store and recall task. The network is asked to reproduce the target trajectory $y_k^{\star t}$ when prompted with a clock-like input signal $I_{\text{clock}}^t = J_{\text{inp}} x_{\text{clock}}^t$, with $J_{\text{inp}} \in \mathbb{R}^{N \times C}$ a random Gaussian matrix with zero mean and $\sigma_{\text{inp}}$ variance and $x_{\text{clock}}^t \in \mathbb{R}^C$, $C = 5$ (see Fig 2A for a graphical representation of the task).

$y_k^{\star t}$ is a temporal pattern composed of $O = 3$ independent continuous signals. Each target signal is specified as the superposition of the four frequencies $f \in \{1, 2, 3, 5\}$ Hz with uniformly extracted random amplitude $A \in [0.5, 2.0]$ and phase $\phi \in [0, 2\pi]$. We defined the additional constraints as described in section 2.2.2 and used the learning rule in Eq (8). Given this formulation, we can arbitrarily modulate the parameters $\mathsf{R}$ and $\tau_\star$.

First we validated the intuition (see Fig 1D and 1E) that larger $\tau_\star$ time constant, with the consequent enrichment of available network states, should progressively erase the difference between an error-based (low-rank) and target-based (high-rank) learning approach. We considered the two scenarios where $\mathsf{R} = 3$ and $\mathsf{R} = N$ and trained the network till convergence for increasing values of $\tau_\star$. The results, collected in Fig 1F, clearly illustrate how the difference between the two approaches vanishes for increasing $\tau_\star$. In Fig 2B and 2C we have reported the output-error (measured as the MSE) and the spike-error $\Delta S = \sum_{it} |s_i^{\star t} - s_i^t|$ as a function of training epochs for a particular choice of the $\tau_\star$ parameter.

Fig 2C shows that for a high rank, the internal activity of the network converges exactly to internal proposed target $s_i^{\star t}$. This confirms that learning with a high rank is equivalent to a

target approach. On the other hand, when the rank is low, this does not happen, and the network autonomously finds an alternative internal dynamics, that is different from $s_i^{*t}$, but still produces an output similar to $y_k^{*t}$ (as shown in Fig 2B).

Both methods achieve low output errors (Fig 2B), with the high-rank approach eventually scoring a lower MSE (the readout limit, i.e. the lowest achievable error given the pre-trained readout matrix $B_{ik}$), while low-rank allows for a faster convergence. To better grasp the interplay between the rank R and the $\tau_\star$ parameters, we trained several instances of the same task to explore the model behavior in the full (R, $\tau_\star$) plane. We measured the output- and spike-error (Fig 2D and 2E), plus an estimate of the convergence time $T_{conv}$ (Fig 2F), quantified as the number of epochs needed to halve the initial output error. Only high-rank feedback achieved low spike-errors (Fig 2E), with a non-trivial dependence on the optimal $\tau_\star$. The LTTS algorithm was found to be the most robust in this sense, reliably achieving low spike errors for a broad range of $\tau_\star$. Interestingly, the MSE metric (Fig 2D) highlighted two regions of low output-error, corresponding either to pure error-based (R = 3) or high-rank, each with different optimal $\tau_\star$. Finally, the convergence time highlights how a low rank systematically allows for a faster convergence (Fig 2F, light region in the bottom-right part of the panel). A possible explanation for this is that, while the target-based (high rank) solution is unique, there are many possible error-based (low-rank) solutions. For this reason is easier to find a close error-based solution starting from a random initial condition in the network activity space (see Fig 1D), while it requires more time to get the target-based solution. However, training slows for low rank and low $\tau_\star$, with magenta-colored conditions signaling failure to reduce output-error by half.

**Dimensionality of the solution space.** The learning formulation of Eq (10) offers a major insights on the role played by the feedback matrix $D_{ik}$. Consider the learning problem (with fixed input and target output) where the synaptic matrix $w_{ij}$ is refined to minimize the output error (by converging to the proper internal dynamics). The learning dynamics can be easily pictured as a trajectory where a single point is a complete history of the network activity $s_n = \{s_i^t : i = 1, \ldots N; t = 1, \ldots T\}_n$, with $n = 1, \ldots E$ where E is the total number of learning epochs. Upon initialization, a network is located at a point $s_0$ marking its untrained spontaneous dynamics. The following point $s_1$ is the activity produced by the network after applying the learning rule defined in Eq (10), and so on. By inspecting Eq (10) one notes that a sufficient condition for halting the learning is $|\sum_i D_{hi}(\bar{s}_i^{*t} - \bar{s}_i^t)| < \epsilon$, where $\epsilon$ is an arbitrary small positive number. If $\epsilon$ is small enough it is possible to write:

$$\sum_i D_{hi}(\bar{s}_i^{*t} - \bar{s}_i^t) \simeq 0.$$

(13)

In the limit of a full-rank D matrix (example: the LTTS limit where D is diagonal) the only solution to Eq (13) is $\bar{s}^t \simeq \bar{s}^{*t}$ and the learning halts only when the target $\bar{s}^{*t}$ is cloned. When the rank is lower, the solution to Eq (13) is not unique, and the dimensionality of possible solutions is defined by the kernel of the matrix D (the collection of vectors $\lambda$ such that $D\lambda = 0$). We have: dim ker D = $N -$ rank D = $N -$ R. We run the store and recall experiment in order to confirm our theoretical predictions.

We repeated the experiment for different values of the rank R. The matrix D is set to $D_{ik} = \frac{\delta_{ik}}{\sqrt{R}}$, $i = 1, \ldots N$, $k = 1, \ldots R$, where $\delta_{ik}$ is the Kronecker delta (the analysis for the case $D_{ik}$ random provides analogous results and is reported in Fig C in S1 Text). When the rank is $N$, different replicas of the learning (different initialization of recurrent weights) converge almost to the same internal dynamics $s_i^t$. This is reported in Fig 3A (left) where a single trajectory represents the first 2 principal components (PC) of the vector $\sum_t |s_i^{*t} - s_i^t|$. The convergence to the

point (0, 0) represents the convergence of the dynamics to $s_i^{*t}$. When the rank is lower (R = $N - 5$, see Fig 3A, right) different realizations of the learning converge to different points, distributed on an line in the PC space. This can be generalized by investigating the dimension of the convergence space as a function of the rank. The dimension of the vector $s_i^{*t} - s_i^t$ evaluated in the trained network is estimated as $d = \frac{1}{\sum_k \lambda_k^2}$, where $\lambda_k$ are the principal component variances normalized to one ($\sum_k \lambda_k = 1$). We found a monotonic relation between the dimension of the convergence space and the rank (see Fig 3B, more information on the PC analysis and the estimation of the dimensionality in Section B in S1 Text). This observation confirms that when the rank is very high, the solution is strongly constrained, while when the rank becomes lower, the internal solution is free to move in a subspace of possible solutions. We suggest that this measure can be used in biological data to estimate the dimensionality of the learning constraints in biological neural network from the dimensionality of the solution space.

**Dimensionality estimation on single trial.**   The dimensionality estimation described above requires the knowledge of the $s^*$ and the repetition of several realizations of the training procedure. However, this information is not available in an experimental setup. For this reason, we propose an alternative approach which could be directly applied to experimental data.

We perform only one realization of the training, but we assume the presence of noise on the synaptic dynamics as follows, by adding white noise to Eq (10) and following the equation:

$$w_{ij}^{t+1} = w_{ij}^t + \Delta w_{ij}^t + \epsilon \xi_{ij}^t \tag{14}$$

where $\epsilon = 0.1$ and $\xi_{ij}^t$ is a normal variable. In Fig 3C we reported dynamics along training epochs of the $\Delta S_i = \sum_t |s_i^{\infty,t} - s_i^t|$ in the first two principal components. Since, the $s_i^{*t}$ is not experimentally accessible, we replaced it with $s_i^{\infty t}$, the internal dynamics of the network at the end of the training. We observe that a first phase is dominated by the learning dynamics, while the second phase is dominated by the synaptic noise. This second phase allows exploring the space of possible internal solutions, even without running several times the training experiment. By estimating the dimensionality of this sampled space (in the same way as described in the previous section) we observe a monotonic dependence between the rank of the matrix D, and the dimensionality of such a space (see Fig 3D). This methodology could be directly applied to data, allowing to provide an estimation of the dimensionality of the space of possible internal solutions to the same problem. We suggest that this could be directly related to the structure of the feedback during training, as demonstrated in our model.

## Application to closed-loop tasks: Behavioral cloning

We face the general problem of an agent interacting with an environment with the purpose to solve a specific task. This is in general formulated in term of an association, at each time $t$, between a state defined by the vector $x_h^t$ and actions defined by the vector $y_k^t$. The agent evaluates its current state and decides an action through a policy $\pi(\{y_k^{t+1}\}|\{x_h^t\})$. Two possible and opposite strategies to approach the problem to learn an optimal policy are Reinforcement Learning and Imitation Learning. In the former, the agent starts by trial and error and the most successful behaviors are potentiated. In the latter the optimal policy is learned by observing an expert which already knows a solution to the problem. Behavioral Cloning belongs to the category of Imitation Learning and its scope is to learn to reproduce a set of expert behaviours (actions) $y_k^{*t+1} \in \mathbb{R}$, $k = 1, \ldots O$ (where $O$ is the output dimension) given a set of states $x_h^{*t} \in \mathbb{R}$, $h = 1, \ldots I$ (where $I$ is the input dimension). Our approach is to explore the implementation of Behavioral Cloning in recurrent spiking networks.

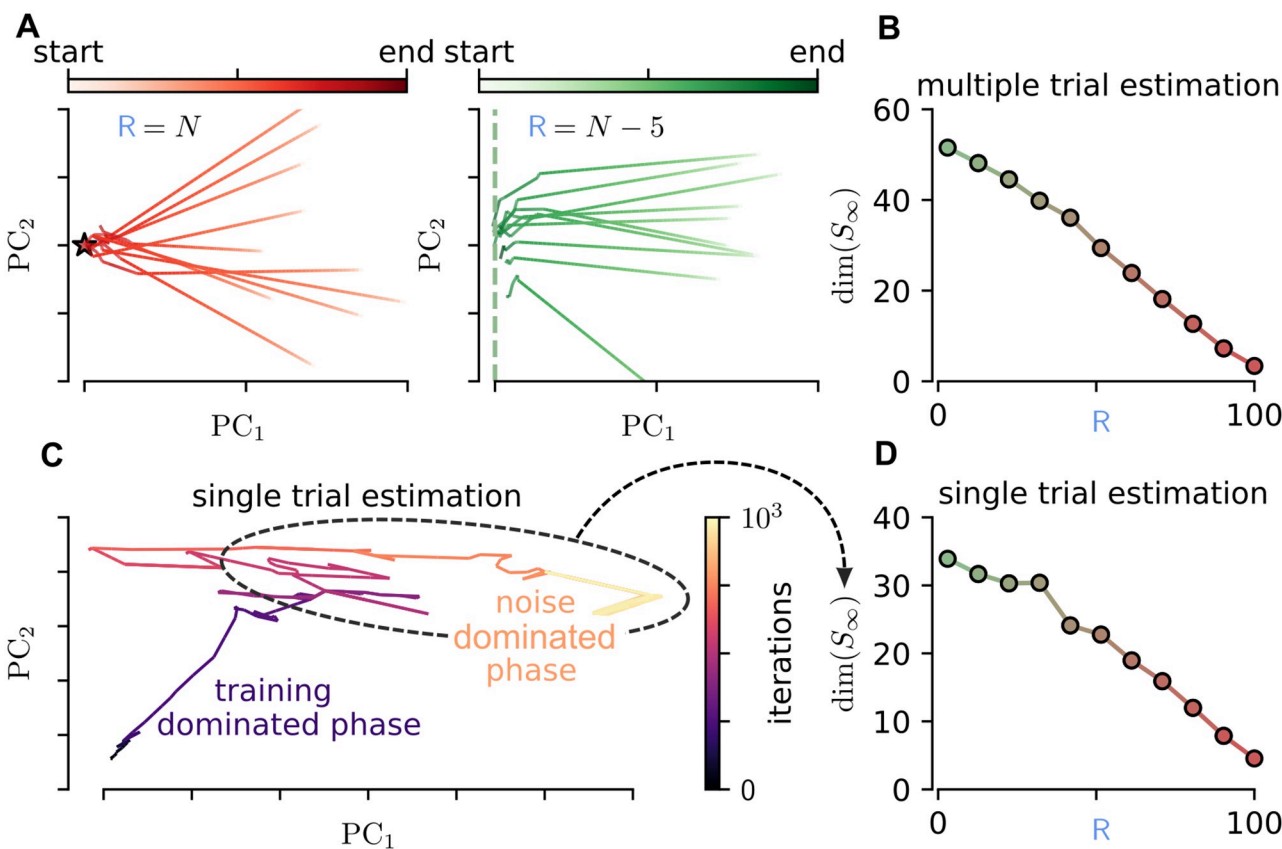

**Fig 3. Error propagation and dimensionality of the internal solution.** (**A**) Dynamics along training epochs of the $\Delta S_i = \sum_t |s_i^{\star t} - s_i^t|$ in the first two principal components for different repetition of the training with variable initial conditions. The error propagation matrix has maximum rank ($\mathsf{R} = N$, target-based limit). (**B**) Same as in (**A**), but with an error propagation matrix with rank $\mathsf{R} = N - 5$. (**C**) Dimensionality of the solution space $\mathcal{S}_\infty$ as a function of the rank $\mathsf{R}$ of the error propagation matrix. (**D**) Dynamics along training epochs of the $\Delta S_i = \sum_t |s_i^{\infty t} - s_i^t|$ in the first two principal components, when a white noise is included in the synaptic dynamics. (**E**) Estimation of the dimensionality of the solution space, sampled thanks to fluctuations induced on the synaptic weights, as a function of the rank $\mathsf{R}$ of the error propagation matrix.

In what follows, we assume that the action of the agent at time $t$, $y_k^t$ is evaluated by a recurrent spiking network and can be decoded through a linear readout $y_k^t = \sum_i \mathsf{B}_{ki} \bar{s}_i^t$, where $\mathsf{B}_{ki} \in \mathbb{R}$. $\bar{s}_i^t$ is a temporal filtering of the spikes $s_i^t$ (similarly to $\hat{s}_i^t$ in Eq (1), with a time scale $\tau_\star$). The network is trained to reproduce the target behavior of the expert $y_k^{\star t}$.

**Button-and-food task.** To investigate the effects of the rank $\mathsf{R}$ of feedback matrix, we design a button-and-food task (see Fig 4A for a graphical representation), which requires for a precise trajectory and to retain the memory of the past states. In this task, the agent starts at the center of the scene, which features also a button and an initially locked target (the food). The agent task is to first push the button so to unlock the food and then reach for it. We stress that to change its spatial target from the button to the food, the agents has to remember that it already pressed the button (the button state is not provided as an input to the network during the task). In our experiment we kept the position of the button (expressed in polar coordinates) fixed at $r_{\mathrm{btn}} = 0.2$, $\theta_{\mathrm{btn}} = -90°$ for all conditions, while food position had $r_{\mathrm{food}} = 0.7$ and variable $\theta_{\mathrm{food}} \in [30°, 150°]$. The agent learns via observations of a collection of experts behaviours, which we indicate via the food positions $\{\theta_{\mathrm{food}}^\star\}$. The expert behavior is a trajectory which reaches the button and then the food in straight lines ($T = 80$). The network receives as

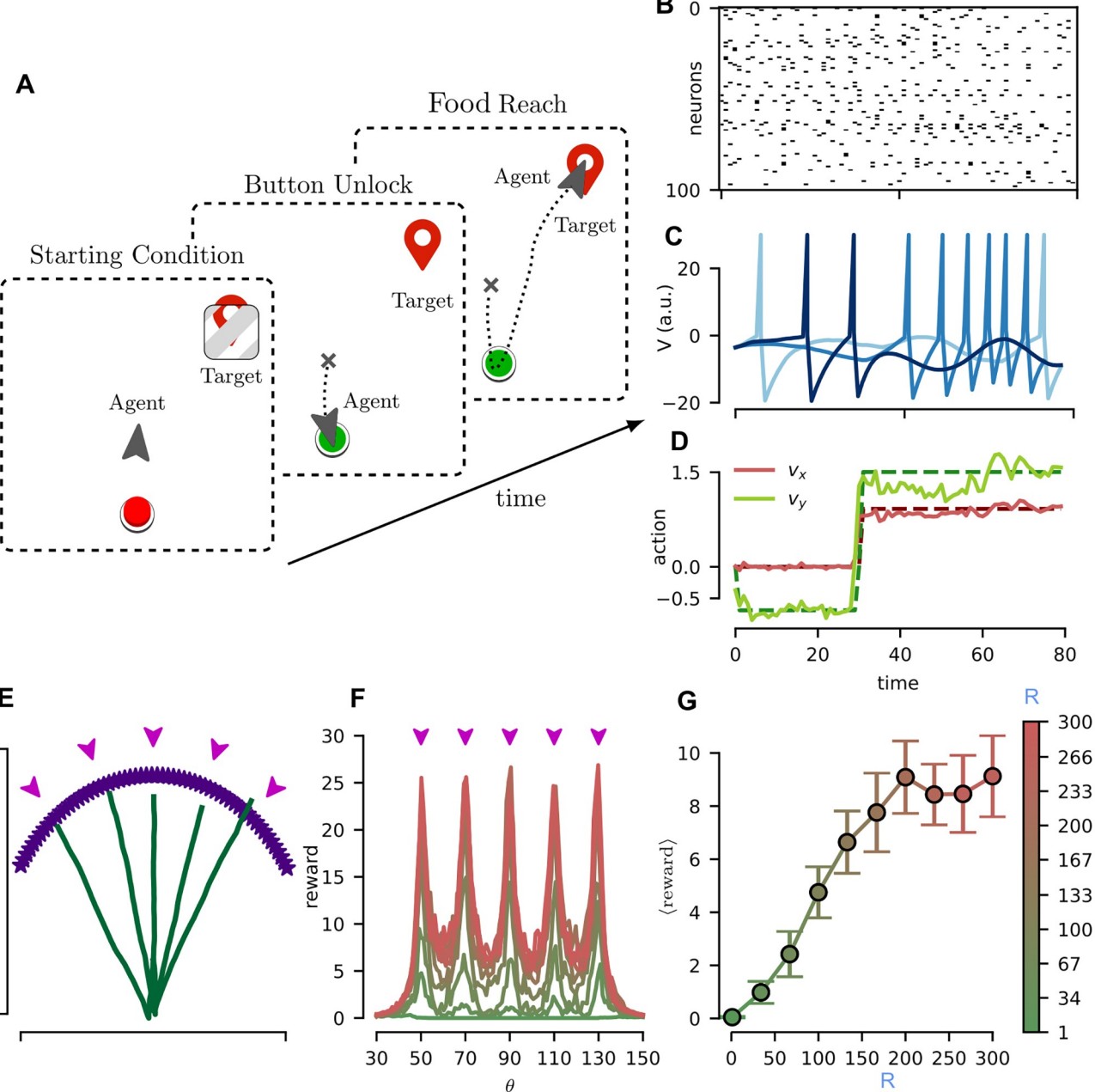

**Fig 4. Button-and-food task.** (**A**) Sketch of the task. An agent starts at the center of the environment domain (left) and is asked to reach a target. The target is initially "locked". The agent must unlock the target by pushing a button (middle) placed behind and then reach for the target (right). (**B**). Rasterplot of the activity of a random sample of 100 neurons across 80 time unit of a task episode. (**C**). Temporal dynamics of the membrane potential of three example units. (**D**) Target $v_{x,y}$ (dashed lines), the velocity direction in the bi-dimensional plane, and the one reproduced by the network after the behavioral cloning (solid lines). (**E**) Example trajectories produced by a trained agent for different target locations. Purple arrows depict the positions of the food for the observed expert behaviors. (**F**) Final reward obtained by a trained agent as a function of the target position (measured by the angle $\theta$ with a fixed radius of $r = 0.7$ as measured from the agent starting position). Individual lines are average values over 10 repetitions. Color codes for different ranks in the error propagation matrix. (**G**) Average over all the target positions of final reward as a function of the rank. Error bars represent the standard deviation of the mean.

input ($I = 80$ input units) the vertical and horizontal differences of both the button's and food's positions with respect to agent location ($\Delta^t = \{\Delta x_b^t, \Delta y_b^t, \Delta x_f^t, \Delta y_f^t\}$ respectively). These quantities are encoded through a set of tuning curves. Each of the $\Delta_i$ values are encoded by 20 input units with different Gaussian activation functions. Agent output is the velocity vector $v_{x,y}$ ($O = 2$ output units). We used $\eta = \eta_{RO} = 0.01$ (with Adam optimizer), moreover $\tau_{RO} = 10$ms. Agent performances are measured by defining a reward function $r$ that considers the importance to push the button before taking the food:

$$r = \frac{\Xi_{\blacksquare}}{\min_t d\left(\boldsymbol{x}_{\text{agent}}^t, \boldsymbol{x}_{\text{food}}\right)},$$

where $\Xi_{\blacksquare}$ is the button-state indicator variable that is zero when the button is locked and one otherwise, the $\boldsymbol{x}_{(\cdot)}^t$ are the agent and target position vectors and $d(\cdot, \cdot)$ is the standard euclidean distance. We repeated training for different values of the rank of the feedback matrix $\mathsf{D}$, computed from $\mathsf{D}_{ik} = \frac{\delta_{ik}}{\sqrt{\mathsf{R}}}$ (with $\delta_{ik}$ the Kronecker delta, the analysis for the case $\mathsf{D}_{ik}$ random provides analogous results and is reported in Section C.2 in S1 Text), in a network of $N = 300$ neurons, and compared the overall performances (more information in Section C.2 in S1 Text). Fig 4B and 4C reports the rastergram for 100 random neurons and the dynamics of the membrane potential for 3 random neurons during a task episode. In Fig 4D we reported an example of the actions ($v_x$, $v_y$, red an green respectively) trajectories, the target ones, and the ones reproduced by the network (dashed and solid lines respectively). In Fig 4E we report the agent training trajectories, color-coded for the final reward and the rewards obtained by the network after the behavioral cloning. Indeed, all the training conditions ($\theta_{\text{food}}^\star \in \{50, 70, 90, 110, 130\}$) show good convergence. In Fig 4F the final reward is reported as a function of the target angle $\theta_{\text{food}}$ for different ranks (ranks are color-coded using the same scheme as Fig 4G and purple arrows indicate the training conditions). As expected, the reward is maximum concurrently with the training condition. Moreover, it can be readily seen how high-rank feedback structures allows for superior performances for this task. Finally, in Fig 4G the average reward across all target conditions is reported as a function of the rank $\mathsf{R}$, further highlighting the benefits of a high-rank feedback structure for this task.

**2D Bipedal Walker.**   We benchmarked our behavioral cloning learning protocol on the 2D Bipedal Walker standard task provided through the OpenAI gym (https://gym.openai.com [38], MIT License). The environment and the task are sketched in Fig 5A: a bipedal agent has to learn to walk and to travel as long a distance as possible. The expert behavior is obtained by training a standard feed-forward network with PPO (proximal policy approximation [39], in particular we used the code provided in [40], MIT License). The sequence of states-actions is collected in the vectors $y_k^{\star t}$, $k = 1, \ldots O$, $x_h^{\star t}$, $h = 1, \ldots I$, $t = 1, \ldots T$, with $T = 400$, $O = 4$, $I = 14$ (we excluded the LIDAR inputs, see Fig 5C for an example of the states-actions trajectories). The average reward performed by the expert is $\langle r \rangle_{exp} \simeq 180$ while a random agent achieves $\langle r \rangle_{rnd} \simeq -120$. We performed behavioral cloning by using the learning rule in Eq (10) in a network of $N = 500$ neurons. We chose the maximum rank ($\mathsf{R} = N$) and evaluate the performances for different values of $\tau_\star$ (more information in Section D in S1 Text). Fig 5B and 5C (bottom) reports the rastergram for 100 random neurons and the dynamics of the membrane potential for 3 random neurons during a task episode. For each value of $\tau_\star$ we performed 10 independent realizations of the experiment. For each realization, the $s_i^{\star t}$ is computed, and the recurrent weights are trained by using Eq (10). The optimization is performed using gradient ascent and a learning rate $\eta = 1.0$. In Fig 5D we report the spike error $\Delta S = \sum_{it} |s_i^{\star t} - s_i^t|$ at the end of the training. The internal dynamics $s_i^t$ almost perfectly reproduces the target pattern of

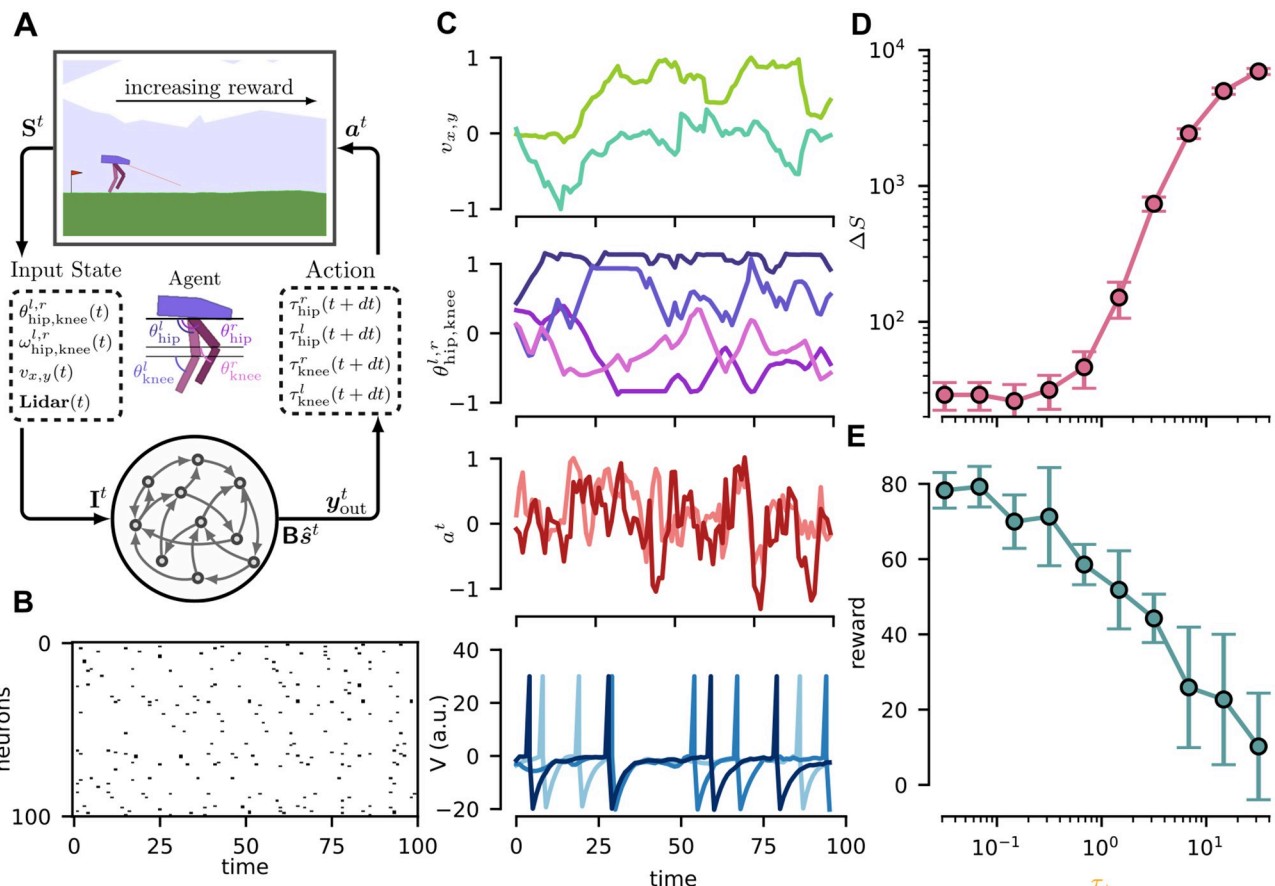

**Fig 5. 2D Bipedal Walker.** (**A**). Representation of the 2D Bipedal Walker environment. The task is to successfully control the bipedal locomotion of the agent, reward is measured as the travelled distance across the horizontal direction. The agent receives a state vector containing several measurements such as joints position, velocity and LIDAR for environment sensing and outputs the torque vector for the four leg joints. (**B**) Rasterplot of the activity of a random sample of 100 neurons across $T = 100$ time unit of a task episode (**C**) Temporal dynamics of a subset of the core input state variables, action vector and spike dynamics. Top panels report respectively: $v_{x,y}$, the velocity vector in the bi-dimensional plane, $\theta^{l,r}_{\text{hip,knee}}$ the angles of the two leg joints with colors matching the scheme of panel **A**, and the action vector $a^t$ containing the torque $\tau_{\text{hip,knee}}$ for the two joints of the left leg. (Bottom) Temporal dynamics of membrane potential for three randomly sampled neurons. (**D**). Average spike error $\Delta S$ as a function of the $\tau_\star$. Error bars represent the standard deviation of the mean. (**E**). Average final episode reward as a function of the $\tau_\star$. Error bars represent the standard error.

spikes $s_i^{\star t}$ for $\tau_\star < 0.5$ms, while the error increases for larger values. The readout time-scale is fixed to $\tau_{\text{RO}} = 5$ms while the readout weights are initialized to zero and the learning rate is set to $\eta_{\text{RO}} = 0.01$. Every 75 training iterations of the readout we test the network and evaluate the average reward $\langle r \rangle$ over 50 repetitions of the task. We then evaluate the maximum $\langle r \rangle$ obtained for each episode (and average it over the 10 realizations). In Fig 5E it is reported the average of the maximum reward as a function of $\tau_\star$. The decreasing monotonic trend suggests that learning with specific pattern of spikes ($\tau_\star \to 0$) enables for optimal performances in this walking task. We stress that in this experiment we used a clamped version of the learning rule. In other words, we substituted $s_i^{\star t}$ to $s_i^t$ in the evaluation of $\frac{\partial v_i}{\partial w_{ij}}$ in Eq (7). This choice, which is only possible when the maximum rank is considered (R = N), allows for faster convergence and better performances. The results for the non-clamped version of the learning rule are reported in section D.2 in S1 Text.

## Discussion

Despite the experimental, theoretical, and computational progresses, neuroscience is still a relatively young field of study. The sign of this can be observed in the fragmented panorama of different theories and models proposed in the literature. In recent years, theoretical neuroscientists have formulated new frameworks attempting at providing more general explanations to aspects concerning intelligence and learning [41, 42]. In this work we contribute to this generalization effort by providing a general framework that is capable to account for different learning approaches by modulating two sensible parameters, the rank of the feedback error propagation $R$ and the tolerance to precise spike timing $\tau_\star$ (see Fig 1C).

We argue that many proposed learning rules can be seen as specific cases of our general framework (e-prop, LTTS, full-FORCE). In particular, the generalization on the rank of the feedback matrix allowed us to understand the target-based approaches as emerging from error-based ones when the number of independent constraints is high. Moreover, we understood that different $R$ values lead to different dimensionality of the solution space. If we see the learning as a trajectory in the space of internal dynamics, when the rank $R$ is maximum, every training converges to the same point in this space. On the other hand, when the $R$ is lower, the solution is not unique, and the possible solutions are distributed in a subspace whose dimensionality is inversely proportional to the rank of the feedback matrix. We suggest that this finding can be used to produce experimental observable to deduce the actual structure of error propagation in the different regions of the brain. On a technological level, our approach offers a strategy to clone on a (spiking) chip an expert behavior either previously learned via standard reinforcement learning algorithms or acquired from a human agent. Our formalism can be directly applied to train an agent to solve closed-loop tasks through a behavioral cloning approach. This allowed solving tasks that are relevant in the reinforcement learning framework by using a recurrent spiking network, a problem that has been faced successfully only by a very small number of studies [3]. Moreover, our general framework, encompassing different learning formulations, allowed us to investigate what learning method is optimal to solve a specific task. We demonstrated that a high number of constraints can be exploited to obtain better performances in a task in which it was required to retain a memory of the internal state for a long time (the state of the button in the button-and-food task). On the other hand, we found that a typical motor task (the 2D Bipedal Walker) strongly benefits from precise timing coding, which is probably due to the necessity to master fine movement controls to achieve optimal performances. In this case, a high rank in the error propagation matrix is not really relevant. From the biological point of view, we conjecture that different brain areas might be located in different positions in the plane presented in Fig 1C.

### Limitations of the study

We chose relevant but very simple tasks in order to test the performances of our model and understand its properties. However, it is very important to demonstrate if this approach can be successfully applied to more complex tasks, e.g. requiring both long-term memory and fine motor skills. It would be of interest to measure what are the optimal values for both the rank of feedback matrix and $\tau_\star$ in a more demanding task. Finally, we suggested that our framework allows for inferring the error propagation structure. However, we observe that the measure we proposed is indirect since it is necessary to estimate the dimensionality of the solution space first and then deduce the dimensionality of the learning constraints. Future development of the theory might be to formulate a method that directly infers from the data the laws of the dynamics in the solution space induced by learning.

## Supporting information

**S1 Text. Supporting Information document.**
(PDF)

## Author Contributions

**Conceptualization:** Cristiano Capone.

**Data curation:** Cristiano Capone, Paolo Muratore.

**Formal analysis:** Cristiano Capone, Paolo Muratore, Pier Stanislao Paolucci.

**Funding acquisition:** Pier Stanislao Paolucci.

**Investigation:** Cristiano Capone, Paolo Muratore.

**Methodology:** Cristiano Capone, Paolo Muratore.

**Project administration:** Cristiano Capone.

**Resources:** Cristiano Capone, Paolo Muratore, Pier Stanislao Paolucci.

**Software:** Cristiano Capone, Paolo Muratore.

**Supervision:** Cristiano Capone.

**Validation:** Cristiano Capone, Paolo Muratore.

**Visualization:** Cristiano Capone, Paolo Muratore.

**Writing – original draft:** Cristiano Capone, Paolo Muratore, Pier Stanislao Paolucci.

**Writing – review & editing:** Cristiano Capone, Paolo Muratore, Pier Stanislao Paolucci.

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
