## [Decision Letter · Decision Letter 0]

6 Jan 2022

Dear Mr. Muratore,

Thank you very much for submitting your manuscript "Error-based or target-based? A unifying framework for learning in recurrent spiking networks" for consideration at PLOS Computational Biology.

As with all papers reviewed by the journal, your manuscript was reviewed by members of the editorial board and by several independent reviewers. In light of the reviews (below this email), we would like to invite the resubmission of a significantly-revised version that takes into account the reviewers' comments.

We cannot make any decision about publication until we have seen the revised manuscript and your response to the reviewers' comments. Your revised manuscript is also likely to be sent to reviewers for further evaluation.

Sincerely,

Michele Migliore

Associate Editor

PLOS Computational Biology

Daniele Marinazzo

Deputy Editor

PLOS Computational Biology

Reviewer's Responses to Questions

**Comments to the Authors:**

Reviewer #1: The authors propose an interesting model to unify target and error-based learning rules.

The proposed framework defines a spectrum of learning regimes thanks to the manipulation of the number of constraints imposed during training.

Such a number of constraints is directly defined thanks to the rank of the error propagation matrix.

The idea underlying the work has the potential to constitute a relevant contribution to the field.

However, the manuscript should be restructured to be clearer and more simulations are needed.

Major:

- The first main concern regards the tasks adopted to demonstrate the advantages of the framework.

For the first task, a high rank of the error propagation matrix seems to be beneficial. For the second application, the importance of the rank is not shown and the authors state that it is almost irrelevant.

Thus, the work demonstrates the advantage of a target-based learning rule, and a successful example of the framework when the system is closer to an error-based regime is missing.

The lack of this demonstration negatively impacts the applicability of the framework and the understanding of the reader.

At the end of the paper, the reader is left wondering why the model, whose aim is to provide a unification between the two learning paradigms,

has been exploited only in the target-based regime.

Thus, I suggest demonstrating some advantages of the system when the rank of the matrix does not correspond to the number of nodes.

If the latter is not achievable, the authors should clearly explain why their model can still offer advantages in comparison to other published alternatives.

For instance, this could be accomplished by comparing the performance of the model proposed with e-prop, LTTS, or full-FORCE.

The convergence time of the model could be also reported for the different cases explored, providing an additional metric to evaluate the performance in the different regimes.

- Relative to the above, it would also be important to show what is the impact of the additional constraints Y_{k, init}. A number of targets equal to N can be set by simply using the untrained network, and

it would be beneficial to know what is the value of O (Eq. 10) in the different cases.

- The paper is difficult to read and should be structured differently. In general, the paper feels fragmented and multiple sections could be merged.

The authors should introduce the neuronal model and the network architecture used, and then introduce the learning rule adopted.

The additional constraints, which are fundamental for the work, are briefly explained in 2.1.5, while other targets are introduced

in Section 2.1.4 by using an untrained network with the desired output as input similarly to the full-FORCE algorithm. The complete set of constraints is unclear.

I would suggest explaining the overall model in one section and drawing an explicit diagram that depicts also the untrained network, the different targets, and where these are coming from.

After such an overall definition, the authors could describe the different learning regimes. Also, it is important to describe how the system is initialized since the untrained network is used for learning.

Minor:

- Fig.1 D should be explained in more detail since it provides information on how the model fits in the literature.

- The notation needs to be unified across the paper. For instance, before the constraints are called q* and then later the authors have introduced Y_{k, init}.

- The statement "The first..." at the end of Section 2.1.5 is unclear.

- Section 2.2.2 and 2.2.1 could be merged, and the title of section 2.2.2 removed. I suggest explaining the second "noisy" methodology to estimate the dimensionality of the solution as an alternative, and to give minor emphasis to

biological data. In the end, biological data have not been used in the paper. Instead, the authors could introduce this alternative approach and its advantages (since repetitions are not needed) and conclude with a note

on biological plausibility.

- Fig. 3 is almost obvious and could go to Supplementary Material. In contrast, Fig. 1 D shows how different models adopt diverse characteristic times. The authors could comment on this and provide some understanding of the relation

between timescale and model. Are the target-based models more flexible? Is the framework proposed more flexible to changes in the characteristic time?

Reviewer #2: This an interesting study that makes a substantial contribution to the topic of learning in the context of network dynamics. The authors study the differences between error-based and target-based approaches, both analytically and numerically, also applying their approach to two ‘real-world’ tasks. I believe that this work can be of interest to a relatively broad readership, such as the network science community, or researchers studying learning, to name a few. Nevertheless, I still have a few small comments for small improvements that I go over below. Also, although the text is understandable, I would advise to have the manuscript proofread by a native English speaker before publication. Likewise, abbreviations used should be initially defined (e.g. the use of ‘LTTS’).

Introduction:

As this paper is aimed at a relatively broad and not specific audience, I think it would be helpful for the reader if the topic was introduced a bit better. For instance, I don’t understand what this sentence comes to say

“When first confronted with reality, humans learn with high sample efficiency, benefiting from the fabric of society and its abundance of experts in all relevant domains.”

Are you referring to babies in this case? If so, do they benefit from experts in different domains? Or do you simply refer to its’ parents?

“Moreover, we observe that spike-timing-based neural codes are experimentally suggested to be important in several brain systems”

You mention ‘we observe’ - perhaps this is a minor detail, but you go on to cite other papers, so this isn’t your observation. Also, I think it would be helpful to explain what ‘spike-timing-based neural codes’ are and why they are important. The connection between this part and the rest of the intro isn’t clear and it comes a bit out of nowhere. I think it should be a paragraph on its’ own, with an explanation of the motivation for this section of the paper.

Lastly, before we move on to the results section, it might be valuable to prepare the reader for what comes up next. You will explore these questions theoretically and then numerically (each with several separate steps). A clarification of the logical steps throughout could be useful.

Results:

As mentioned, there are a few small language issues. For instance in section 2.1.3

“propose an alternative formulations allowing to evaluate target”

Should be corrected.

I think that the paper could either be re-structured, or, at least, that the authors would make better connections between the different sections. For instance, in section 2.2.3 you start by saying:

“As we discussed above the τ⋆”

Where is this discussed above? It isn’t anywhere in section 2.2.2. Why is this here now?

In the experiment section, many choices are made by the authors. Although this is of course necessary, no explanation for the reasoning behind these choices are given (e.g.:

“The readout time-scale is fixed to τRO = 5ms while the readout weights are initialized to zero and the learning rate is set to ηRO = 0.01.”

Why are these values chosen? I think there should be some kind of ‘sensitivity analysis’, showing that similar results would be gained with slight variations of these parameters, or, at least, giving justification for them.

Discussion:

The discussion is a bit disappointing, as it is very brief and doesn’t really engage with the literature. If this work isn’t placed adequately within the existing studies, it can loose much of its relevance. Also, there is no discussion with regards to the limitations of this study.

On a more specific note (and perhaps this is a bit outside the usual scope of the field), I think that it would be interesting to consider, and discuss, the consequence of non-experts in the vicinity of the agent. In this paper, you assume that the agent learns from experts that surround it. This is a fair starting assumption, but in reality we usually have heterogenous populations where actually there is a chance of learning undesirable traits (by imitation).

Figure 3:

I think that in subplots A and B it would be better to name the x-axis ‘number of iterations’.

The caption of Panel C reads “Scatter plot of mse vs ΔS for different values of τ⋆.” But on the plot itself we have the absolute value of ΔS. This should be resolved. Also, this isn’t a scatter plot. This is also color-coded, but compared with plots A and B, this isn’t mentioned. As the color coding is equivalent in all three, this can be mentioned once for the whole figure.

Figure 4:

Subplot C isn’t helpful in its’ current form. Either show less examples, colour-code them differently (keeping everything blue makes it harder to distinguish), or expand the size of the plot.

For subplot D please write explicitly what the dashed and solid lines are.

In Panel G - what do the error bars show? Please state this explicitly. Also, what is the color gradient? Same as in F? If so it should be placed on the right of G, or G should have its own.

Figure 5:

Subplot B should be split into two (as it has two subplots). So the bottom one should have its own letter referring to it.

Subplot D has the same problem as Fig4C.

Reviwers conclusion:

In conclusion, I think this is a nice and insightful piece of work, and I would therefore recommend it for publication after minor revision. As the changes suggested are minor, the editor may decide to accept them without my further involvement. But I am also happy to check the revised version of this manuscript.

Reviewer #3: The authors introduced a general framework for supervised learning in recurrent spiking networks, with two main parameters, the rank of the feedback error propagation and the tolerance to precise spike timing. And they analyzed the learning performance for different parameters. The research topic is very important for not only constructing efficient machine learning methods but also understanding biological neural networks. However, I have some concerns below.

Major comments

- While error-based learning is familiar, target-based learning is a bit unfamiliar to many readers. Therefore, I recommend that the authors add more explanations about target-based learning in the Introduction. Also, they should describe the critical differences in the characteristics and learning performances between the two. Moreover, LTTS and full-Force, which appear in Results, should be explained briefly.

- The authors state that ‘Moreover, our general framework, encompassing different learning formulations, allowed us to investigate what learning method is optimal to solve a specific task.’ in the Discussion. However, only two tasks, the button-and-food task and the 2D Bipedal Walker, were used for checking the performance in the manuscript. It seems that the best performance was achieved at a higher rank of error propagation matrices. What kind of task favors a lower rank of error propagation matrices? After all, the higher the rank of D, the better it is? If so, the claim ‘different brain areas might be located in different positions in the plane presented in Fig.1C’ is invalid for the rank of D (error-based or target-based learning).

Minor comments

- Clarify the definition of \\delta_t and \\tau_star in Eq. (2).

- Why is the star mark in \\tau of Eq.(2)? It’s confusing to me because it looks like a desired time scale parameter.

- Fig. 4F: We cannot see the error bars in F, but there is a description of error bars in the caption.

- Fig. 4G: We can see the messy graph around 300.

- In p.9, Fig.5D, Fig.5G -> Fig.4D, Fig.4G

- What’s the reward? The performance r?

**Have the authors made all data and (if applicable) computational code underlying the findings in their manuscript fully available?**

Reviewer #1: Yes

Reviewer #2: Yes

Reviewer #3: Yes

PLOS authors have the option to publish the peer review history of their article (what does this mean?). If published, this will include your full peer review and any attached files.

Reviewer #1: No

Reviewer #2: No

Reviewer #3: No
---

## [Decision Letter · Decision Letter 1]

4 Apr 2022

Dear Mr. Muratore,

Thank you very much for submitting your manuscript "Error-based or target-based? A unifying framework for learning in recurrent spiking networks" for consideration at PLOS Computational Biology. As with all papers reviewed by the journal, your manuscript was reviewed by members of the editorial board and by several independent reviewers. The reviewers appreciated the attention to an important topic. Based on the reviews, we are likely to accept this manuscript for publication, providing that you modify the manuscript according to the recommendations of Rev.1.

Sincerely,

Michele Migliore

Associate Editor

PLOS Computational Biology

Daniele Marinazzo

Deputy Editor

PLOS Computational Biology

[LINK]

Reviewer's Responses to Questions

**Comments to the Authors:**

Reviewer #1: The manuscript has been improved considerably.

However, we feel that it is still necessary to improve the clarity of exposition.

Practically, the authors propose an interesting analysis and methodology to expand the target dimensionality to control the number of constraints that the network is subjected to. The expansion of the target is random and accomplished by adding random values in the definition of the 'read-out' matrix. The fact that this expansion can help

the learning process to discover a better solution is interesting per se. We feel that this result should be clearly introduced as one of the main contributions of the work.

The idea is clearly fascinating and the analysis is novel, but it is still not simple to understand the complete training process.

We advise the authors to describe, point by point, how learning is accomplished.

After careful reading:

- First, the target signal has been processed by the untrained network.

- Then, the read-out has been trained to reproduce the target signal.

- The output connectivity matrix is expanded to include possible constraints.

- Finally, such an output connectivity matrix is adopted in the learning process to adapt the recurrency.

We think it would be better to describe this methodology clearly in one section and to justify the different choices.

Since the core of the work lies in the possibility to add constraints, thanks to which it is possible to move in the target/error-based spectrum, we suggest that the authors should introduce the different tasks by saying explicitly what the number of constraints vs the dimensionality of the target is. This can be inferred and it was explicit in your

previous response, but saying it also in the text would help the reader.

We have also some comments on the figures:

- Fig. 1D-E. How is this figure generated? The authors should clarify if this is an intuitive explanation of the different regimes of learning or if it was generated quantitatively.

The regime where tau/delta_t<<1 would correspond to a simulation where the discretization step is greater than the characteristic time, and I am unsure how this could be

practically achieved with numerical methods. Can you clarify this point?

Also, the ideal space represented has \\delta s_1 and \\delta s_2 as x and y-axis, but the line corresponding to the error-based regime is increasing until it reaches the target-based limit for a maximum rank. In that limit, \\delta s\\approx 0. I would have imagined the target-based limit on the left approximately at zero, and the error-based limit on the right. Increasing the rank would correspond to going from right to left in this case, etc. Probably, the intended ideal x-axis is the rank.

Also, what are the circles? Do they simply reflect the density of the possible solutions? I would avoid writing 'good' or 'bad'.

I feel that some clarifications are needed, but these panels are surely an interesting addition to the paper.

- Fig.2 C I can understand that the target-based limit recovers the correct succession of spikes more accurately than the error-based regime, but I am not sure that in this way the result is meaningful. Simply, the error-based regime is characterized by a low-rank matrix and can not find the correct multi-dimensional succession of spikes. If I understood correctly, the result is an immediate consequence of the methodology adopted. Perhaps, it would be interesting to see if the error-based regime can recover the succession of spikes from the 'original' target, neglecting the additional constraints.

Minor:

The text has typos, please do extensive proofreading.

Fig.2 A, please consider using dots for the target to make it visible.

Fig.2 D-E-F, please fix the labels on the x-axis.

Reviewer #2: the authors have replied to my comments and I therefore recommend the paper for publication.

Reviewer #3: The authors have done a great job in addressing my concerns in the revision of the manuscript.

**Have the authors made all data and (if applicable) computational code underlying the findings in their manuscript fully available?**

Reviewer #1: None

Reviewer #2: Yes

Reviewer #3: None

PLOS authors have the option to publish the peer review history of their article (what does this mean?). If published, this will include your full peer review and any attached files.

Reviewer #1: No

Reviewer #2: No

Reviewer #3: No

Figure Files:

Data Requirements:

Reproducibility:

References:

---

## [Editor Report · Decision Letter 2]

17 May 2022

Dear Mr. Muratore,

We are pleased to inform you that your manuscript 'Error-based or target-based? A unifying framework for learning in recurrent spiking networks' has been provisionally accepted for publication in PLOS Computational Biology.

Best regards,

Michele Migliore

Associate Editor

PLOS Computational Biology

Daniele Marinazzo

Deputy Editor

PLOS Computational Biology

---

## [Editor Report · Acceptance letter]

15 Jun 2022

PCOMPBIOL-D-21-02197R2 

Error-based or target-based? A unified framework for learning in recurrent
spiking networks

Dear Dr Muratore,

I am pleased to inform you that your manuscript has been formally accepted for publication in PLOS Computational Biology. Your manuscript is now with our production department and you will be notified of the publication date in due course.

With kind regards,

Olena Szabo
